# Development of New Extracts of *Crocus sativus* L. By-Product from Two Different Italian Regions as New Potential Active Ingredient in Cosmetic Formulations

**Maria Rosa Gigliobianco** [1,†], **Manuela Cortese** [1,†], **Dolores Vargas Peregrina** [1,2], **Carla Villa** [3], **Giulio Lupidi** [1], **Letizia Pruccoli** [4], **Cristina Angeloni** [1], **Andrea Tarozzi** [4], **Roberta Censi** [1,2] **and Piera Di Martino** [1,2,*]

1 School of Pharmacy, Università di Camerino, 62032 Camerino, Italy; maria.gigliobianco@unicam.it (M.R.G.); manuela.cortese@unicam.it (M.C.); dolores.vargas@unicam.it (D.V.P.); giulio.lupidi@unicam.it (G.L.); cristina.angeloni@unicam.it (C.A.); roberta.censi@unicam.it (R.C.)

2 Recusol srl, 62032 Camerino, Italy

3 Department of Pharmacy, Università di Genova, 16126 Genova, Italy; carla.villa@unige.it

4 Department for Life Quality Studies, University of Bologna, 40126 Bologna, Italy; letizia.pruccoli2@unibo.it (L.P.); andrea.tarozzi@unibo.it (A.T.)

* Correspondence: piera.dimartino@unicam.it; Tel.: +39-07-3740-2215; Fax: +39-07-3763-7345

† Authors equally contribute to the manuscript.

**Abstract:** This project aimed to apply eco-friendly extraction methods to *Crocus sativus* L. by-product (flowers without stigmas i.e., tepals composed of petals and sepals) to recover extracts with high antioxidant capacity and polyphenol content, to be used in cosmetic products. Flowers grown in two different Italian regions (Sample 1—Alba in Piemonte, north of Italy and Sample 2—Sibillini in Marche, centre of Italy) were subjected for the first time to different eco-friendly microwave-mediated green solvents extractions (MGSE) andquali-quantitative determination in antioxidant molecules. Firstly, the extracts from Sample 1 were selected according to their total phenol content (TPC) by Folin–Ciocalteu's assay and antioxidant capacity (AC) by spectrophotometric assays. Then, according to preliminary results, MGSE carried out in ethanol 70°, water, and glycerin were selected as the most performing methods and applied to both Samples 1 and 2. The best results were obtained using green solvents, such as water or ethanol 70°, for the samples coming from Marche. The identification and quantification of phenolic compounds, belonging to anthocyanins and flavonols classes, was performed by using UPLC-DAD-ESI-MS. Concerning flavonols content, the most abundant analyte is kaempferol 3-*O*-sophoroside and the extract in water from Sample 1 showed the higher amount of flavonols, reaching the concentration of 25.35 mg of kaempferol 3-O-glucoside equivalent per gram of tepals DW of raw material. Among anthocyanins, the most abundant was delphinidin 3,5-O-diglucoside and the high concentration of anthocyanin was detected in water and ethanol extract. Two new compounds, myricetin-di-glucoside and primflasine, were identified for the first time in *Crocus sativus* L. by-product by high-resolution mass spectrometry (HRMS). The green batches obtained by extraction were thus characterized and evaluated for their biological potential and safety in keratinocyte HaCaT cells. The extracts were not cytotoxic up to 0.03 mg/mL. The water and ethanol 70° extracts were the most effective in counteracting oxidative stress induced by H2O2 and UVA exposure and reduced cytotoxicity induced by UVB exposure. The water extract was also able to significantly reduce cytotoxicity induced by sodium dodecyl sulphate-induced damage. Taken together, these results suggest a potential use of these waste materials as cosmeceutical preparations such as antiaging, and as anti-skin irritation formulation by-products.

**Keywords:** *Crocus sativus* L. wastes; eco-friendly extractions method; UPLC-DAD-ESI-MS method; HRMS analysis; cytotoxicity; oxidative stress; cosmetic products

## 1. Introduction

Interest in the development of sustainable processes for the production or extraction of bioactive compounds from agro-food organic waste has recently increased due to the potential applications of these compounds in food, cosmetics, and pharmaceuticals [1,2].

The spice saffron, with its unique aroma, color, and flavor, consists of the stigmas of *Crocus sativus* L. (Iridaceae). Flowers without stigmas, and more specifically tepals which are mainly composed of petals indistinguishable from sepals are currently considered as waste material (1 kg of spice from saffron stigmas generates 150,000 blooms and 1500 kg of leaves) [3].

The *Crocus sativus* L. (Iridaceae) is cultivated notably in Western Asia, with Iran as the world's largest producer, but is also of significant economic importance to large parts of Mediterranean Europe (Greece, Spain), particularly for the traditional plain of Navelli in Italy [4,5]. Nonetheless there are several other regions in Italy were the *Crocus sativus* L. is started to be cultivated [6], in particular the Sibillini area in the Marche region due to a project named "Programma Leader plus 2000~2006" based on the exploitation of small farms that through the production of this spice they might have the possibility to reflect from the commercial point of view the quality of the area. In our study, the by-product of *Crocus sativus* L. were compared for the first time with the ones cultivated in Alba (Piemonte region). In collaboration with local farmers and cosmetic laboratories of Marche and Piemonte, an experimental research was started to exploit by-product from *Crocus sativus* L. to be used as potential active ingredients for cosmetic formulations.

Our interest in the *Crocus sativus* L. by-product is due to the fact that, similarly to stigmas [7–9], tepals are already known as a source of antioxidants such as flavonols, flavanones, and apocarotenoids [10], responsible for a variety of health-enhancing properties and could potentially be used as functional components for pharmaceuticals, nutraceuticals, and cosmetic formulations [11].Particularly, many extracts based on flowers of *Crocus sativus* L. are proposed and are commercially available as a valid supplement of antiaging, anti-inflammatory, antimicrobial, and antioxidant formulations. Natalia Moratalla-López et al. reported that tepals extracts, a promising source of antioxidants, have been used as ingredients in high-end cosmetic products to develop new dermal treatments, since they could contrast the process of aging [12]. So far, several phenolic compounds have been identified, such as benzoic acids, hydroxycinnamic acids [13], anthocyanins, and flavonols [14]. Among anthocyanins, delphinidin 3,5-di-*O*-glucoside is the most abundant one, followed by petunidin 3,7-di-*O*-glucoside, petunidin 3-*O*-glucoside and malvidin 3-*O*-glucoside [14–16]. Flavonols are mainly represented by kaempferol, quercetin, and isorhamnetin glycosides [14–19].

To ensure the quality and the quantity of these extracted active ingredients, it is necessary to take into account several elements like environmental conditions, plant growth, harvesting time, but also the process steps as plant material desiccation, extraction, separation, and purification. Among these steps, one of most important is the extraction. Several methods were developed to optimize the extractions of *Crocus sativus* L. by-products, such as the conventional solid-liquid extraction (CSLE), ultrasound-assisted extraction (UAE) and microwave-assisted extraction (MAE) [19], natural deep eutectic solvent as L-lactic acid/glycine (5:1) [20], and cold-pressed saffron (*Crocus sativus* L.) floral by-product [10].

Among all, an attractive green technique is the microwave-mediated green solvents extraction (MGSE), which was thus selected for the present work, where it was associate with the natural deep eutectic solvent (NADES).

Actually, microwave extraction method, using microwave energy, induces molecular motion by ionic conduction and dipole rotation. With this extraction procedure, it is possible to increase temperature and pressure generating changes in the cell structure and improving the penetration of the selected solvent across the sample matrix [21]. One of the most important advantages of microwave extraction is based on its activation energy directly on the target molecules, that help to improve the extraction efficiency to optimize the timing of the entire process [22]. Besides, this extraction method, when compared with

other conventional extractive procedures, presents many other advantages: reduction of time, solvents amount and energy consumption; higher extraction rates, greater purity of the final product, and better products quality with lower costs [23–25]. Da Porto and Natolino [19] compared the microwave assisted extraction method of saffron floral bioresidues powder with other extraction methods such as the conventional solid-liquid and the ultra-sound assisted extractions. The efficiency of the microwave-assisted extraction method was assessed by keeping constant both the extraction solvent (ethanol/water mixture, 59:41 *v/v*) and the solvent extraction temperature (66 °C), and by changing solid to liquid ratios (1:10, 1:20, 1:30 and 1:50 g mL$^{-1}$) and extraction times (0.5, 1, 2, 3, 4 and 5 min). The authors kinetically tested extracts for total phenol content, total anthocyanins, and antioxidant activity through spectrophotometric methods. by-product.

In our study, MGSE was tested in presence of different green extraction solvents such as water, ethanol 70°, ethanol 96°, and glycerine, but also natural deep eutectic solvents, that are defined as green solvents based on the synthesis of natural substrate, such as sugars, organic acids, amino acids, etc., that can be used in a green process for the extraction of bioactive compounds [26]. In our study, the selection of the most promising solvents was preliminary performed according to the total phenol content and the antioxidant capacity results determined through spectrophotometric methods. Then, the UPLC-DAD-ESI-MS method permitted the identification and quantification of the most abundant polyphenols in the selected extracts (water, ethanol 70°, and glycerine), and high-resolution mass spectrometry (HRMS) was helpful for the identification of two compounds detected in saffron by-product for the first time. The different extracts coming from Marche and Alba were tested in vitro on human keratinocyte cell line, HaCaT, to assess their cytoprotective activity against UVB irradiation, oxidative stress induced by hydrogen peroxide (H2O2) and UVA exposure, and irritation triggered by the detergent sodium dodecyl sulphate (SDS).

## 2. Materials and Methods

### 2.1. Chemicals

1,1-Diphenyl-2-picrylhydrazyl (DPPH), 2,4,6-Tris(2-pyridyl)-s-triazine (TPTZ), (±)-6-Hydroxy-2,5,7,8-tetramethylchromane-2-carboxylic acid (TROLOX), 2,2′-Azino-bis(3-ethylbenzothiazoline-6-sulfonic acid) diammonium salt (98%TLC) (ABTS, gallic acid, sodium carbonate monohydrate ACS reagent, 2′-7′ dichlorodihydrofluorescein diacetate (DCFH$_2$-DA), 3-(4,5-dimethyl-2-thiazolyl)-2,5-diphenyl-2H-tetrazolium bromide (MTT), H$_2$O$_2$, SDS, sodium acetate and ethanol (ethanol absolute grade) were purchased from Sigma-Aldrich (Stenheim, Germany). Manganese (IV) oxidize activated (≥90%), and Folin Ciocalteu's phenol reagents were purchased from Fluka (Buchs, Switzerland). Anhydrous sodium acetate, Iron (III) anhydrous hydrochloride were purchased from J.T. Baker Analyzed (Center Valley, PA, USA) and sodium carbonate anhydrous was purchased from Carlo Erba (Milano). All solvents and reagents were of analytical grade. Pure standard kaempferol 3-O-glucoside and delphinidin 3-O-β-D-glucoside chloride we obtained from Primary Reference Standard, HWI Group (Rheinzaberner, Germania) and Sigma-Aldrich (Stenheim, Germany) respectively. Formic acid and acetonitrile for LC/MS were purchased from Carlo Erba Reagents (Cornaredo, MI, Italy). All the other chemicals were provided by Sigma Aldrich (Stenheim, Germany).

### 2.2. Plant Materials

Two different *Crocus sativus* L. tepals were used for the study: Sample A: Piemonte, Alba, October 2017 and Sample C from Marche Region, October 2017.

During the flowering period, they were manually picked, and stigmas were separated from the rest of the plant. Plants were immediately desiccated in a ventilated oven (VEC2103/8, Everest, Rimini, Italy) until constant weight at 37 °C and then used for extraction.

### 2.3. Extraction Procedures

Extractions were carried out by MGSE in a microwave applicator, consisting in a prototype multimode cavity, equipped with a Pyrex™ reactor, a magnetron operating at 2.45 GHz, two optical fibers for the temperature measurement, and a control unit, which allows to manage and modulate different parameters of the process, such as emitted power, time, and temperature. Extractions were performed in the presence of glycerine (G), water (H), or ethanol 70° (EH). Accurately weighted plants (4.5 g) were added with solvent to a volume of 20 mL and then placed in the Pyrex™ vessel heated under microwave irradiation (70 °C for 30 min.). The resulting extracts were stored in a refrigerator at 4 °C in 50 mL polyethylene vials with a screw cap (BD Falcon™, BD Biosciences, Bedford, MA, USA).

### 2.4. Total Phenol Content Determination

Total Phenol Content (TPC) was determined according to the Folin-Ciocalteu spectrophotometric method by measuring the absorbance at 765 nm [20,21]. Results were expressed as milligrams of gallic acid equivalents (GAE) per grams of by-product (mg GAE/g).

### 2.5. Evaluation of the Antioxidant Capacity (AC)

Antioxidant activity was evaluated by measuring 1,1-diphenyl-2-picrylhydrazyl (DPPH$^{\bullet}$) radical scavenging activity [22,23], 2,2′-azino-bis (3-ethylbenzothiazoline-6-sulphonic acid) (ABTS$^{\bullet+}$) radical cation scavenging capacity [23,24] and Ferric Reducing Antioxidant Capacity (FRAP) [25,26]. Trolox (6-hydroxy-2,5,7,8-tetramethylchroman-2-carboxylic acid) was used as calibration standard. Values were expressed as $IC_{50}$, defined as the concentration of the tested material required to cause a 50% decrease in initial DPPH, ABTS or iron concentration, as well as µmol Trolox equivalent $(TE)g^{-1}$ of sample.

### 2.6. Ultra Performance Liquid Chromatography—Diode Array-Electrospray Ionization-Mass Spectrometry Analysis

Identification and quantification of polyphenols was performed by a UPLC (Agilent 1290 Infinity Technologies UPLC, Santa Clara, CA, USA) coupled with diode array detector (DAD) and a triple quadrupole mass spectrometer (MS-QQQ) (Series 6420, Agilent Technologies Santa Clara, CA, USA) equipped with an electrospray ionization source (ESI). The UPLC was equipped with a binary pump and auto-sampler. The chromatographic separation was carried out in presence of a Luna (C18 150 × 4.6 mm) column with 5 µm particle diameter (Phenomenex, Castel Maggiore, BO, Italy) and a temperature of 25 °C. The gradient elution was a mobile phase of water/1% formic acid (solvent A) and acetonitrile/1% formic acid (solvent B) at a constant flow of 1 mL/min. The optimized gradient was not linear: 0 min, 10% B; 0–24 min, 44% B. The injection volume was 5 µL.

The DAD was set at wavelengths of 348 and 526 nm for the quantification of flavonols and anthocyanins, respectively. Based on preliminary data recovered during the identification of polyphenols, and according to the literature [14,16], two external standards, kaempferol-3-O-glucoside and delphinidin-3-O-β-D-glucoside chloride, were selected to specifically quantify flavonols and anthocyanins, respectively. The linearity, sensitivity, accuracy, and precision of the developed method were verified according to the Food and Drug Administration Guidelines (FDA) [27], as detailed in the Supplementary Materials. Then, all the analytes present in each sample were quantified and expressed in mg of kaempferol-3-O-glucoside equivalent for flavonols, and for anthocyanins mg of delphinidin-3-O-β-D-glucoside chloride equivalent per g of dry weight (mg $g^{-1}$ DW) [16].

Each sample was extracted in triplicate, individually analysed, and the RSD was calculated for each compound.

For the correct identification of the different signals, the DAD analyses were coupled with mass spectrometer detection in both negative and positive ionization mode. In the negative mode, a large group of compounds was identified corresponding to the deprotonate molecules ions of different flavonols. The anthocyanins were detected in the positive mode. The working conditions of the ESI source were the following: gas temperature

350 °C; gas flow 12 L/min; nebulizer pressure 55psi. Samples (A2 and C2) were diluted with ultrapure water in a 1:2 ratio, then sonicated for 5 min, and centrifuged at 12,000 rpm for 10 min (Scilogex D3024R High-Speed Refrigerated Micro-Centrifuge, Rocky Hill, CT, USA). Solutions were then filtered by a 0.20 μm filter (Captiva Econofilter, PTFE, Agilent Technologies, Santa Clara, CA, USA). The mass spectrometer analysis was performed in full scan mode in $m/z$ 200–820 (negative) and $m/z$ 200–700 (positive). Subsequently, the correspondent precursor ion was subjected to several product ion experiments by changing the collision energy (CE) in the range of 10–40 eV under the following conditions: the MS2 scan range was set at $m/z$ 200–820, the scan time at 250 and 300 ms, the fragment at 20 eV, and the cell accelerated voltage was set at 7 eV. The obtained product ion spectra gave structural information, especially on the aglycone identity and its linked sugar moieties.

### 2.7. Identification of Unknown Molecules by High-Resolution Mass Spectrometry (HRMS)

UPLC-HRMS spectrometry was performed by UPLC (1290 Infinity II) coupled with a mass spectrophotometer with a high-resolution chromatography/electrospray ionization quadrupole time-of-flight tandem mass spectrometry named QTOF-6545 (Agilent technologies 6545, Santa Clara, CA, USA). The calibration of instrument was settled at a low mass range 1700 $m/z$ max, slice mode in high resolution, and the extended dynamic range was 4GHz. The chromatographic separation was carried out in presence of an Eclipse Plus C18 RRHD 2.1 × 50 mm 1.8 μm column (Agilent Technologies, Santa Clara, CA, USA). The analysis was performed in presence of a gradient elution as previously described at a constant flow of 0.2 mL/min. The optimized gradient is not linear: 0 min, 20% B; 0–30 min, 50% B. The injection volume was 1 μL. Sample C2 with a dilution 1:2 was analyzed in both positive and negative ionization mode with full scan acquisition to identify unknown signals. The working conditions of the Q-TOF were the following: gas temperature 130 °C; drying gas 11 L/min; nebulizer 55 psi, sheath gas temperature 350 °C and sheath gas flow 12L/min. UPLC–HRMS was performed in full-scan mode to obtain the precursor ion high resolution mass, and successively targeted MS/MS mode for structural informationwith TOF analyzer in positive and negative ionization polarity. For targeted MS/MS analyses, the CE spread was 10, 20 and 40 eV, accordingly with the ones used for databases creation. The TOF acquisition range was set from 100 to 1000 $m/z$ range (acquisition rate 1 spectrum/s). Data processing was performed with Agilent Mass Hunter Qualitative Analysis software version 10.0. Confidence of compound molecular formula identification was based on accurate mass measurements and isotope pattern and expressed by 'overall identification score' computed as a weighted average of the isotopic pattern signals of a compound, such as exact masses, relative abundances and $m/z$ distances (spacing). The contribution to overall score of parameters were: mass score = 100, isotope abundance = 60, isotope spacing = 50, mass expected data variation 2.0 mDa $^+$ 5.6 ppm, mass isotope abundance 7.5%, MS/MS 5mDa $^+$ 7.5 ppm. Mass Hunter Molecular Structure Correlator (MSC) software was used to calculate the correlations between the MS/MS fragmentation observed and the molecular structures proposed by using two scores, namely the Molecular Formula Generator (MFG) and Molecular Structure Correlator (MSC).

### 2.8. Determination of Cytotoxicity, Cytoprotection and Antioxidant Activity in Keratinocyte Cells
2.8.1. Cell Cultures

Human keratinocyte cell line, HaCaT, were routinely grown in Dulbecco's modified Eagle's Medium (DMEM) supplemented with 10% fetal bovine serum, 2 mM L-glutamine, 50 U/mL penicillin and 50 μg/mL streptomycin at 37 °C in a humidified incubator with 5% $CO_2$. To evaluate cytotoxicity and cytoprotection, HaCaT cells were seeded in 96-well plates at $2 \times 10^4$ cells/well. To determine the intracellular antioxidant activity, HaCaT cells were seeded in 96-well plates at $3 \times 10^4$ cells/well. All experiments were performed after 24 h of incubation at 37 °C in 5% $CO_2$.

### 2.8.2. UVA and UVB Irradiation

Before irradiation, treatment medium was removed from culture plates, and the HaCaT cells were washed twice, before being covered with a thin layer of Hank's Balanced Salt Solution (HBSS). Control cells were treated in the same way but were not irradiated. For UVA irradiation, we used a bank of two Philips TLK 40W/10R fluorescence tubes (Sara srl, Varese, Italy), emitting 350–400 nm energy, peaking at 365 nm. For UVB irradiation, we used a bank of two Philips TL 20W/12 fluorescence tubes (Sara srl, Varese, Italy), emitting 280–320 nm energy with a peak at 312 nm. The fluorescence tubes were positioned to deliver 4.095 mW/cm$^2$ and 0.865 mW/cm$^2$ of UVA and UVB, respectively, as measured by a radiometer (Goldilux, Oriel).

### 2.8.3. Cytotoxicity

Cell viability was evaluated by the reduction of MTT to its insoluble formazan, as previously described [28]. Briefly, HaCaT cells were treated for 24 h with different concentrations of extracts (0.003–3 mg/mL) at 37 °C in 5% $CO_2$. Subsequently, the treatment medium was replaced with MTT in HBSS (0.5 mg/mL) for 2 h at 37 °C in 5% $CO_2$. After washing with HBSS, formazan crystals were dissolved in isopropanol. The amount of formazan was measured (570 nm, reference filter 690 nm) using the multilabel plate reader VICTOR™ X3 (PerkinElmer, Waltham, MA, USA). The cell viability was expressed as percentage of control cells.

### 2.8.4. Cytoprotection

To evaluate the cytoprotective effects of extracts against the cytotoxicity elicited by UVB radiation, HaCaT cells were treated for 1 h with different concentrations of extracts (0.003–0.03 mg/mL) at 37°C in 5% $CO_2$ and then exposed to 50 mJ/cm$^2$ of UVB. After irradiation with UVB, HaCaT cells were washed with *phosphate buffered saline* and then incubated with DMEM 10% FBS in 5% $CO_2$ at 37 °C for 24 h.

The cytoprotective effects of extracts were also evaluated against the cytotoxicity elicited by SDS. HaCaT cells were treated for 2 h with different concentrations of extracts (0.003–0.03 mg/mL) and then treated with 140 μM SDS for 24 h at 37 °C in 5% $CO_2$.

For both experiments, at the end of the treatment, the cell viability was determined by MTT assay (see Section 2.8.3).

### 2.8.5. Intracellular Antioxidant Activity

The antioxidant activity of extracts in terms of ability to counteract oxidative stress was evaluated by the fluorescent probe DCFH$_2$-DA, as previously described [29].

The antioxidant activity was studied against $H_2O_2$ and UVA. HaCaT cells were treated for 2 h (for the evaluation against $H_2O_{2)}$ and 1h (for the evaluation against UVA) with different concentrations of extracts (0.003–0.03 mg/mL) at 37 °C in 5% $CO_2$ then 100 μL of DCFH$_2$-DA (10 μg/mL) was added to each well. After 30 min of incubation at room temperature, DCFH$_2$-DA solution was replaced with $H_2O_2$ (100 μM) for 30 min. or with HBSS for the exposure with 5 J/cm$^2$ UVA (see Section 2.8.2.). The intracellular oxidation of DCFH$_2$ to fluorescent end-product 2′–7′ dichlorofluorescein (DCF) or the ROS formation was measured (excitation at 485 nm and emission at 535 nm) using the multilabel plate reader VICTOR™ X3 (PerkinElmer, Waltham, MA, USA). Data are expressed as fold increase of DCF fluorescence versus untreated cells.

### 2.9. Statistical Analysis

The biological data are shown as mean $\pm$ standard error (SEM) of at least three independent experiments. In this regard, statistical analysis was performed using one-way ANOVA (analysis of variance) with Dunnett or Bonferroni post hoc tests, as appropriate. Analyses were performed using GraphPad PRISM software (version 5.0; GraphPad Software, La Jolla, CA, USA) on a Windows platform.

## 3. Results and Discussion

### 3.1. Determination of the Total Phenol Content and Antioxidant Capacity of Extracts

In Table 1 Total Phenol Content (TPC) and Antioxidant Capacity (AC) of six batches were reported, A1, A2 and A3 (from Alba) and C1, C2 and C3 (from Sibillini) to provide the full antioxidant abilities of the tested extracts. The used solvents, selected as green extraction solvents, were water, ethanol 70°, and glycerine for comparison. The selection of water as elective solvent was carried out as previously suggested [30]. We also chose ethanol and glycerine as comparative solvents. The lowest TPC was exhibited by the glycerin extracts for both samples while the highest one was that from water and then from ethanol 70°. The high value of water extract in terms of TPC were also reported by Menghini and co-authors [31]. The highest values for ABTS, FRAP, and DPPH were observed for the glycerol extracts for both the Alba and Sibillini regions, and the lowest ABTS, FRAP, and DPPH values were those for water extract. Since all these values may be affected by different molecules, an indepth identification and quantitation of the polyphenols, especially the anthocyanis and flavonols contained in the different extracts, is then necessary.

**Table 1.** Total Phenol Content (TPC) and antioxidant capacity (AC) of different extracts.

| Batches | Total Phenol Content (mg GAE/l) | ABTS μmol TEA/mL | FRAP (μmol TEA/mL) | DPPH (μmol TEA/mL) |
|---|---|---|---|---|
| A1 (EtOH/$H_2O$ Alba) | 151.138 ± 15.205 | 411.222 ± 26.352 | 467.118 ± 10.726 | 518.443 ± 16.288 |
| A2 ($H_2O$ Alba) | 1020.951 ± 70.802 | 154.175 ± 4.164 | 419.985 ± 10.811 | 167.229 ± 2.1004 |
| A3 (Gly Alba) | 55.995 ± 0.058 | 888.907 ± 109.053 | 808.927 ± 189.782 | 878.107 ± 10.127 |
| C1 (EtOH/$H_2O$ Sibillini) | 204.571 ± 4.885 | 795.218 ± 20.614 | 488.909 ± 10.556 | 684.315 ± 10.822 |
| C2 ($H_2O$ Sibillini) | 1400.950 ± 13.369 | 114.884 ± 2.312 | 409.029 ± 7.236 | 110.312 ± 7.588 |
| C3 (Gly Sibillini) | 62.485 ± 7.225 | 1040.150 ± 12.604 | 846.637 ± 16.258 | 998.123 ± 10.556 |

### 3.2. Identification of Flavonols and Anthocyanins

3.2.1. Ultra Performance Liquid Chromatography-Tandem Mass Spectrometry (UPLC-MS/MS)

The identification of flavonoids was carried out referring to previous studies [12,16,23,32]. The discussion about the attribution of every peak to specific substances is reported in Supplementary Materials. In our approach, we identified twenty-one flavonols and anthocyanins based on their chemical structure, building up a method to rationally identify all the obtained signals as described in the Supplementary Materials for each detected peak. More specifically, in case of isomeric forms, the position of the sugar moiety was assigned in relation to the check of correct CE from 10 to 30 eV useful for the cleavage of the glycosidic bond, that was reported for the first time. Literature data reported a major lability of the sugar in position 7 with respect to the position 3 of the flavonoid species [14,33]. Based on this approach, at increasing values of CE, it was possible to identify the di-glycoside, tri-glycoside forms of kaempferol, for isorhamnetin glycoside species by the sequential loss of each individual sugar moiety from the aglycone of kaempferol. Moreover, in the case of delphinidin, belonging to anthocyanins species, in literature [10,14] was identified only the delphinidin 3,5-O-diglucoside, while in our case we detected three signals at different CE, based on the different positions of the glucosyl moiety that could be in -3, 5, or 7 position of the aglycone.

We also revealed the presence of two flavonol species F3 (*m/z* 641, RT = 7.38 min.) and F7 (*m/z* 711, RT = 9.01 min.) never found in saffron tepals extracts, according to the literature. The identification of these two compounds was confirmed by HRMS analysis as described in Section 3.2.2.

3.2.2. Identification of Unknown Molecules by High-Resolution Mass Spectrometry (HRMS)

We used the HRMS to identify two unknown molecules that we were not able to definitively assign using low resolution tandem mass spectrometry according to data reported in scientific literature. By the scan analysis, the compound molecular formula was obtained. Meanwhile the targeted MS/MS (the accurate mass of F3 corresponded to the formula $C_{27}H_{30}O_{18}$, in case of F7 corresponded to $C_{31}H_{36}O_{19}$) allowed to investigate the possible molecular structures by using online available databases, such as PubChem and ChemSpider.

In HRMS compound **F3** corresponded to the ion at 643.1507 *m/z* in the positive polarity. The ESI-MS/MS of this precursor ion revealed the formation of two main product ions at *m/z* 481.0983 and *m/z* 319.0453 respectively (Table 2). The product ion at *m/z* 481.0983 corresponded to $[M-162]^+$ correlated to the loss of one glucosyl unit. Instead, the product ion showing almost equal intensity at *m/z* 319.0453, that can be related to $[M-162-162-H]^+$, was formed by the loss of two glucosyl moieties. This fragment can be identified with the aglycone of myricetin. By targeted MS/MS experiment, the spectrum is stored by using different CE (10, 20 and 40 eV). This acquisition mode increases the number of fragments produced, as their formation is related to the power of the CE used. This means that the MS/MS spectrum is full of useful information for structure correlation, and it can be compared with those spectra present in databases and registered in the same conditions. For the molecule identification we used both ChemSpider and PubChem databases. Between the two sources, PubChem gave the higher number of results, proposing 16 structures. Among them, on the base of the fragmentation pattern of flavonols present in the glycosylated forms of myricetin were selected as the most likely proposal. In particular, PubChem Database suggested five isomeric forms of glycoside myricetin. Among them, the structures with a disaccharide sugar moiety, such us the sophoroside unit, gave the higher score (MFG = 46.4, MSC = 77.6; overall = 36.0). Unfortunately, we did not detect the characteristic fragment associated with the sophoroside group, obtained from the neutral loss of 181amu. For all these reasons, we identified the unknown peak F3 as a di-glucoside form of myricetin. The position of the two individual glucoside moieties remains undetermined without the use of an appropriate standard. According to PubChem database, the score associated to each diglucoside myricetin isomer was myricetin 3,4′-diglucoside (MFG = 46.4, MSC = 75.7; overall = 35.1); myricetin 3,5′-diglucoside (MFG = 46.4, MSC = 75.7; overall = 35.1), myricetin 3,5-diglucoside (MFG = 46.4, MSC = 75.7; overall = 35.1) and myricetin 3,7-diglucoside (MFG = 46.4, MSC = 75.7; overall = 35.1). The presence of myricetin in these extracts was confirmed in the literature only by the HPLC-DAD, and no evidence of the identification of myricetin by mass spectrometry and HRMS was pointed out before of this work [34]. In Figure 1, the fragmentation spectra for the HRMS characterization of myricetin di-glucoside is reported.

The compound **F7** was detected both in positive and negative polarity with an accurate mass of *m/z* 713.1930 and *m/z* 711.1776 respectively (Table 2). Moreover, the targeted MS/MS experiment showed the formation of two main product ions at *m/z* 551.1399 and 287.0553 in the positive mode, corresponding to the loss of glucosyl moiety $[M-162+H]^+$ and the mass of aglycone, respectively. In addition, in the negative ionization mode, the precursor ion at *m/z* 711.1776 produced several product ions: at *m/z* 667.1500, associated to the loss of $-C_2H_4O$, at *m/z* 651.1565, correlated with the loss of $C_2H_4O_2$, while the product ion at *m/z* 609.1456 was associated to the loss of $C_4H_6O_3$. Finally, the product ion at *m/z* 284.0325 corresponded to the aglycone portion. The aglycone can be addressed to kaempferol, but the fragmentation pattern of the sugar moiety did not fulfil the criteria of the others flavonols, leading to suppose the presence of an unconventional sugar. In this case, both ChemSpider and PubChem suggested the same list of potential compounds. Among flavonols proposed, the only compounds related to a glycosylated form of kaempferol was primflasine, with a substituted glucoside. The identification score for primflasine, according to ChemSpider were MFG = 99.2, MSC = 76.8; overall = 76.1. So,

we supposed to identify A7 as primflasine, belonging to the large class of flavonols [35]. Figure 2 presents the fragmentation spectra for the HRMS characterization of primflasine.

**Table 2.** Identification of unknown molecules by HRMS analysis.

| Peak Number | R.T. | Ionization Mode | Ions Experimental | Ions Theoretical | Error ppm | Formula | Principal Putative ion | MSC % * ChemSpider | MSC % * PubChem | Identification |
|---|---|---|---|---|---|---|---|---|---|---|
| F3 | 0.895 | Positive | 643.1507 (P) | 643.151 | −0.4665 | $[C_{27}H_{30}O_{18}+H]^+$ | Aglycone-diglucoside | 76.9 | 77.6 | Myricetin-diglucoside |
| | | | 481.0983 (F) | 481.0982 | 0.2079 | | Aglycone-glucoside | 99.4 | 99.4 | |
| | | | 319.0453 (F) | 319.0454 | −0.3134 | | Aglycone | 98.5 | 99.6 | |
| F7 | 1.162 | Positive | 713.1930(P) | 713.1929 | 0.1402 | $[C31H36O19+H]^+$ $[C31H36O19+Na]^+$ | Glycoside | 75.7 | 75.7 | Primflasine |
| | | | 735.1745(A) | 735.1748 | −0.4081 | | Sodium adduct | 97.7 | 97.7 | |
| | | | 551.1399(F) | 551.1395 | 0.7258 | | Aglycone-glucoside | 99.6 | 99.6 | |
| | | | 287.0553(F) | 287.055 | 1.0451 | | Aglycone | | | |
| | | Negative | 711.1776(P) | 711.1778 | −0.2812 | $[C_{31}H_{36}O_{19}]^-$ | Glycoside | 77.9 | 77.9 | |
| | | | 667.1500(F) | 667.1516 | −2.3983 | | $C_2H_4O$ | 93.9 | 93.9 | |
| | | | 651.1565(F) | 651.1567 | −0.3071 | | $C_2H_4O_2$ | 96.7 | 96.7 | |
| | | | 609.1456(F) | 609.1461 | −0.8208 | | $C_4H_6O_3$ | 94.1 | 94.1 | |
| | | | 284.0325(F) | 284.0326 | −0.3521 | | Aglycone | 99.3 | 99.3 | |

P: Precursor ions; F: Fragmentor; A: Adduct; *: The percentage of molecular structure correlator (MSC%) between MS/MS fragments observed and the putative molecular structure.

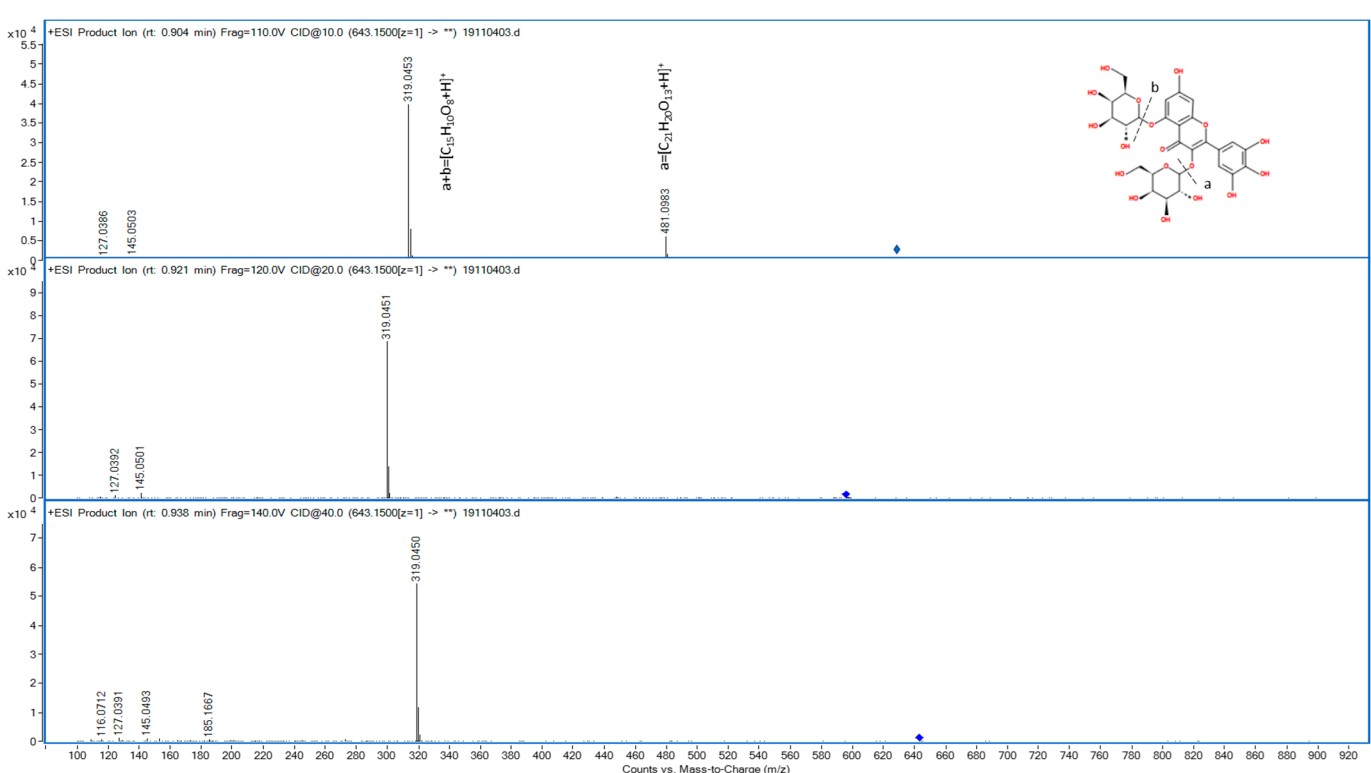

**Figure 1.** High resolution mass spectrum of myricetin diglucoside (643 *m/z*).

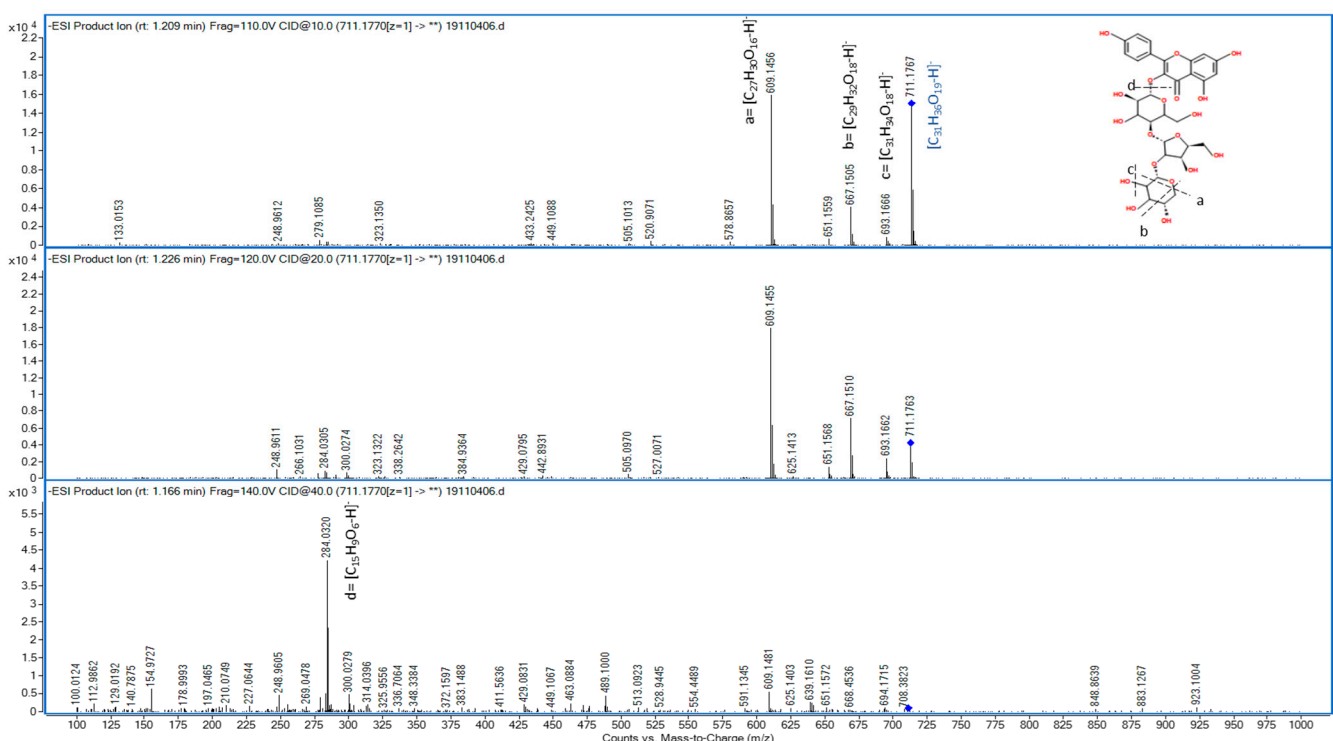

**Figure 2.** High resolution mass spectrum of primflasine (711 *m/z*).

### 3.2.3. Quantitative Analysis by UPLC-DAD-ESI-MS

Results for the quantification of flavonols and anthocyanins in the samples A1, A2, A3, C1, C2, and C3 are reported in Tables 3 and 4. Due to the complexity of the extracts in terms of number of molecules, and the commercial unavailability of standard for each compound, we decided to use one external standard for each class of flavonoid [16]. Reasonably, the number of sugar moieties and their position do not affect the quantification because sugars do not absorb at the wavelength set for the analysis.

**Table 3.** Quantification of flavonols by using UPLC-DAD-ESI-MS expressed as mg of kaempferol 3-O-glucoside equivalent per gram of DW of raw material. Each sample was extracted in triplicate and individually analysed. The standard deviation was calculated on three replicates.

| | A1 | | A2 | | A3 | | C1 | | C2 | | C3 | |
|---|---|---|---|---|---|---|---|---|---|---|---|---|
| **Identification** | **Conc.** [1] | **SD%** | **Conc.** [1] | **SD%** | **Conc.** [1] | **SD%** | **Conc.** [1] | **SD%** | **Conc.** [1] | **SD%** | **Conc.** [1] | **SD%** |
| Kaempferol 3,7, 4′- tri-O-glucoside | 0.24 | 0.1 | 2.18 | 0.3 | 1.48 | 0.1 | 1.99 | 0.2 | 1.37 | 0.9 | 1.35 | 0.1 |
| Kaempferol 3,7-O-diglucoside | 0.03 | 0.3 | 0.29 | 0.1 | 0.17 | 0.1 | 0.15 | 7.1 | 0.20 | 8.9 | 0.12 | 0.1 |
| Myricetin diglucoside | 0.09 | 0.1 | 0.19 | 0.2 | 0.16 | 0.3 | 0.19 | 0.1 | 0.30 | 0.3 | 0.14 | 0.2 |
| Quercetin 3,7-O-diglucoside | 0.02 | 0.4 | 0.22 | 0.4 | 0.12 | 1.7 | 0.12 | 6.3 | 0.17 | 0.2 | 0.10 | 0.1 |
| Isorhamnetin 3,7-O-diglucoside | 0.07 | 0.5 | 0.53 | 0.3 | 0.53 | 1.9 | 0.70 | 1.3 | 0.77 | 4.3 | 0.63 | 0.1 |
| Quercetin 3-O-sophoroside | 0.69 | 0.1 | 1.36 | 0.1 | 1.32 | 0.2 | 1.22 | 0.5 | 1.46 | 1.1 | 0.96 | 0.1 |
| Primfrasine | 0.12 | 0.3 | 0.72 | 0.1 | 0.56 | 0.4 | 0.57 | 1.0 | 0.75 | 0.9 | 0.51 | 0.1 |

**Table 3.** *Cont.*

| Identification | A1 | | A2 | | A3 | | C1 | | C2 | | C3 | |
|---|---|---|---|---|---|---|---|---|---|---|---|---|
| | Conc.[1] | SD% | Conc.[1] | SD% | Conc.[1] | SD% | Conc.[1] | SD% | Conc.[1] | SD% | Conc.[1] | SD% |
| Kaempferol 3-*O*-sophoroside | 12.84 | 0.1 | 17.53 | 0.1 | 17.16 | 0.2 | 18.10 | 0.1 | 21.91 | 0.2 | 12.11 | 0.1 |
| Kaempferol 3-*O*-rutinoside | 0.04 | 0.1 | 0.08 | 0.2 | 0.06 | 10.5 | 0.07 | 1.7 | 0.11 | 3.7 | 0.05 | 0.1 |
| Isorhamnetin 3-*O*-rutinoside | 0.25 | 0.2 | 0.46 | 0.2 | 0.65 | 1.0 | 0.42 | 0.1 | 0.57 | 0.4 | 0.38 | 0.1 |
| Quercetin 3-*O*-glucoside | 0.08 | 0.2 | 0.23 | 0.1 | 0.17 | 0.1 | 0.26 | 0.2 | 0.43 | 0.1 | 0.22 | 0.3 |
| Kaempferol 3-*O*-(6″-acetyl-glucoside) 7-*O*-glycoside | 0.01 | 2.9 | 0.01 | 0.1 | 0.01 | 0.1 | 0.01 | 1.9 | 0.02 | 8.3 | 0.01 | 0.1 |
| Kaempferol 3-glucoside | 0.12 | 0.7 | 0.41 | 2.6 | 0.29 | 0.6 | 0.14 | 0.5 | 0.03 | 0.5 | 0.07 | 1.0 |
| Isorhamnetin 3-*O*-glucoside | 0.33 | 0.1 | 0.74 | 0.1 | 0.54 | 0.1 | 1.24 | 0.1 | 1.45 | 0.1 | 0.98 | 0.1 |
| Kaempferol 3-sophoroside-7-rhamnoside | 0.09 | 0.1 | 0.18 | 0.3 | 0.06 | 1.3 | 0.14 | 1.0 | 0.25 | 1.3 | 0.13 | 0.8 |
| Kaempferol 3-*O*-(6″-acetyl-galactoside) or Kaempferol 3-*O*-(6″-acetyl-glucoside) | 0.03 | 0.2 | 0.08 | 1.9 | 0.04 | 2.7 | 0.05 | 0.2 | 0.10 | 6.4 | 0.04 | 0.7 |
| Kaempferol 3-*O*-(6″-acetyl-galactoside) or Kaempferol 3-*O*-(6″-acetyl-glucoside) | 0.03 | 7.1 | 0.06 | 0.7 | 0.04 | 0.9 | 0.02 | 0.1 | 0.01 | 2.2 | 0.01 | 1.8 |
| Quercetin 3-*O*-glucoside-7-*O*-rhamnoside | 0.02 | 0.8 | 0.04 | 0.2 | 0.05 | 2.1 | 0.10 | 0.1 | 0.05 | 0.1 | 0.08 | 0.2 |
| Kaempferol 3-*O*-glucoside-7-*O*-rhamnoside | <LOQ | - | 0.01 | 10.7 | 0.01 | 0.1 | 0.03 | 1.4 | 0.01 | 2.4 | 0.02 | 5.3 |
| Isorhamnetin 3-*O*-glucoside. 7-*O*-rhamnoside | 0.04 | 1.2 | 0.01 | 1.4 | 0.01 | 0.5 | 0.02 | 1.3 | 0.01 | 0.1 | 0.02 | 6.0 |
| Kaempferol | 0.11 | 2.8 | 0.03 | 7.0 | 0.13 | 6.1 | 0.10 | 5.8 | 0.28 | 1.7 | 0.07 | 9.1 |
| **Total amount** | **15.23** | | **25.35** | | **23.56** | | **25.61** | | **30.26** | | **17.98** | |

[1] Concentration expressed as mg of kaempferol 3-O-glucoside equivalent per g of DW of raw material.

Concerning flavonols, the most abundant analyte is kaempferol 3-*O*-sophoroside, followed by quercetin 3-*O*-sophoroside, isorhamnetin 3-*O*-glucoside, and kaempferol 3,7,4′-*O*-triglucoside. The high content of kaempferol glycosides is in accordance with the results reported in literature and detected in the petals of *CS* [16]. The aqueous extracts of both samples C2 and A2 showed the highest concentration in flavonols. The total amount of flavonols in sample C2 was 30.26 mg of kaempferol 3-O-glucoside equivalent per gram of DW of raw material. The high amount of flavonols was detect in Alba in sample A2, reaching the concentration of 25.35 mg of kaempferol 3-O-glucoside equivalent per gram of tepals DW of raw material and in sample of Camerino in C1 with a concentration of

25.61 mg of kaempferol 3-O-glucoside equivalent per gram of tepals DW of raw material. Regarding anthocyanin contents, they are much less present with respect to flavonols in all extracts. Among them the most abundant analyte, also in accordance with the literature [10,14], is the delphinidin 3,5-O-diglucoside, followed by petunidin 3,5-O-diglucoside. The higher number of anthocyanins was detected in samples C1 and A2. The lowest concentration of six anthocyanins identified was found in the samples C3 and A1. The total content of anthocyanins in C1 was 1.86 mg of delphinidin 3-O-glucoside equivalent per gram of DW of raw material, meanwhile in the sample A2 their concentration was 1.18 mg of delphinidin 3-O-glucoside equivalent per gram of DW of raw material. The different contents in terms of total amount of flavonols and anthocyanins observed in the samples of Alba and Camerino should be attributed to the peculiar climatic and geological conditions of these two areas, which are important aspects investigated for other *Crocus Sativus* L. cultivation in Italy [36].

**Table 4.** Quantification of anthocyanins by using UPLC-DAD-ESI-MS expressed as mg of delphinidin 3-O-glucoside equivalent per gram of DW of raw material. Each sample was extracted in triplicate and individually analyzed. The standard deviation was calculated on three replicates.

| | A1 | | A2 | | A3 | | C1 | | C2 | | C3 | |
|---|---|---|---|---|---|---|---|---|---|---|---|---|
| Identification | Conc. [1] | SD% | Conc. [1] | SD% | Conc. [1] | SD% | Conc. [1] | SD% | Conc. [1] | SD% | Conc. [1] | SD% |
| Delphinidin 3,5-O-diglucoside | 0.25 | 0.4 | 0.98 | 0.4 | 0.84 | 0.2 | 1.26 | 0.7 | 1.11 | 1.6 | 0.73 | 0.6 |
| Petunidin 3,5-O-diglucoside | 0.03 | 5.0 | 0.16 | 0.3 | 0.11 | 0.1 | 0.21 | 0.7 | 0.17 | 2.1 | 0.11 | 0.8 |
| Delphinidin 3-O-β-D-glucoside | 0.03 | 0.9 | 0.04 | 0.3 | 0.05 | 0.4 | 0.13 | 1.3 | 0.04 | 3.3 | 0.02 | 2.1 |
| Malvidin diglucoside | <LOQ | 4.0 | 0.01 | 0.8 | 0.01 | 0.1 | 0.03 | 1.2 | 0.01 | 9.4 | 0.01 | 5.7 |
| Delphinidin diglucoside | <LOQ | - | <LOQ | - | <LOQ | - | <LOQ | - | <LOQ | - | <LOQ | - |
| Delphinidin diglucoside | <LOQ | - | <LOQ | - | <LOQ | - | <LOQ | - | <LOQ | - | <LOQ | - |
| **Total amount** | **0.35** | | **1.18** | | **1.00** | | **1.86** | | **1.33** | | **0.86** | |

[1] Concentration expressed as mg of delphinidin 3-O-glucoside equivalent per g of DW of raw material.

### 3.3. Antioxidant and Cytoprotective Activity of Sibillini Extracts in Human Keratinocytes

As the extracts from the Sibillini region C1, C2, and C3 showed the highest content of total phenols, flavonols, and anthocyanins, as well as the highest total phenol content compared to the extracts from Alba. This is why they were selected to carried out the biological assays. The cytotoxicity was evaluated in keratinocyte HaCaT cells with extract concentrations ranging from 0.003 to 3 mg/mL for 24 h and cell viability was evaluated by MTT assay. The treatment of HaCaT cells with the extracts at concentrations lower than 0.3 mg/mL did not affect cell viability (Figure 3). The range of concentrations 0.003–0.03 mg/mL was therefore selected for the subsequent experiments. Under physiological conditions, skin cells, such as keratinocytes, are characterized by oxidative stress mainly due to the exposure to the UVA and UVB radiation in natural sunlight [37,38]. The effects of skin exposure to UV radiation or chemical compounds are well known to induce skin aging, as well as the promotion of irritation and several diseases, such as skin cancers [39]. In particular, UVB radiation with high energy exerts cytotoxic and mutagenic effects while UVA radiation with low energy promotes [40] oxidative stress, inflammation, and several photoaging processes [41]. To study the antioxidant activity of the extracts, in terms of ability to counteract oxidative stress, HaCaT cells were exposed to both $H_2O_2$ treatment and UVA radiation. In particular, HaCaT cells were exposed to 100 $\mu$M $H_2O_2$ for 30 min and 5 J/cm$^2$ UVA after a treatment of 2 h and 1 h, respectively, with the studied extracts (0.003 and 0.03 mg/mL). At the end of the pro-oxidant exposure, intracellular oxidative stress

was measured by probe $H_2DCF$-DA. As shown in Figure 4, the treatment of HaCaT cells with C1 and C2 extracts significantly reduced oxidative stress triggered by $H_2O_2$ at 0.003 and 0.03 mg/mL. Meanwhile, C3 extract did not influence oxidative stress. Remarkably, C1 and C2, but not C3, extracts were also able to significantly counteract oxidative stress induced by UVA irradiation at both concentrations (Figure 5).

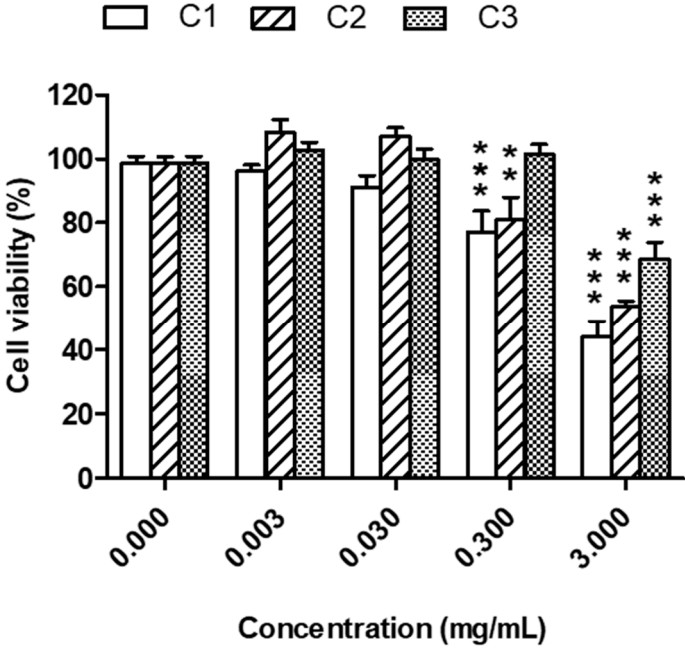

**Figure 3.** Cytotoxicity of C1, C2 and C3 extracts in HaCaT cells. Cells were treated with various concentrations of extract (0.003–3 mg/mL) for 24 h. At the end of treatment, cell viability was evaluated by MTT assay, as described in the method section. Data are expressed as percentage of control cells and expressed as mean $\pm$ SEM of four independent experiments (** $p < 0.01$ and *** $p < 0.001$ vs. cells untreated; one-way ANOVA with Dunnett post hoc test).

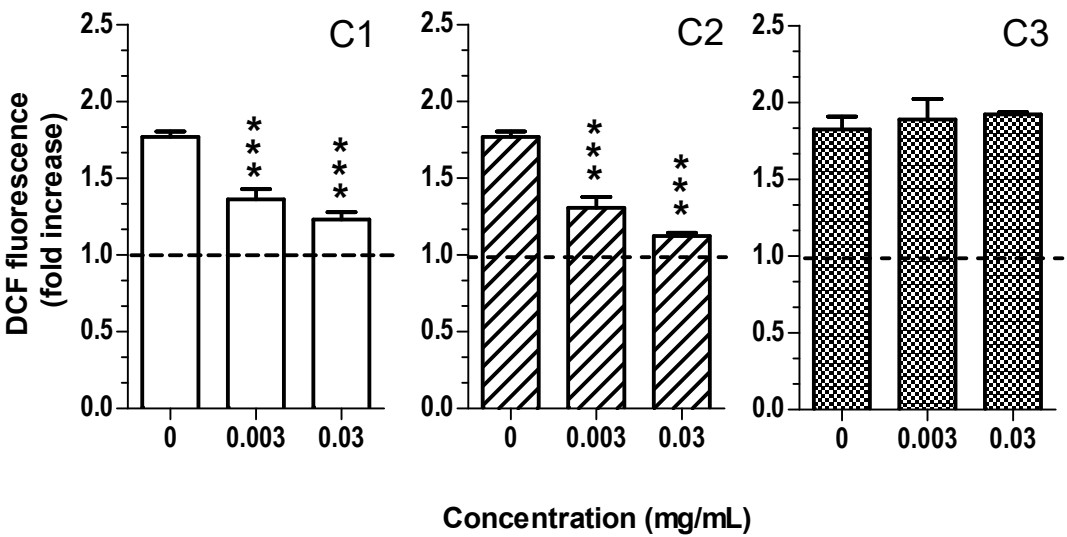

**Figure 4.** C1 and C2, but not C3, extracts reduce oxidative stress induced by $H_2O_2$ in HaCaT cells. Cells were treated with various concentrations of extract (0.003–0.03 mg/mL) for 2 h and then treated with $H_2O_2$ (100 µM) for 30 min. At the end of treatment, intracellular oxidative stress was evaluated using the probe $DCFH_2$-DA, as described in the method section. Data are expressed as fold increase of fluorescent DCF formation versus untreated cells and reported as mean $\pm$ SEM of three independent experiments (*** $p < 0.001$ vs. cells treated with $H_2O_2$; one-way ANOVA with Dunnett post hoc test).

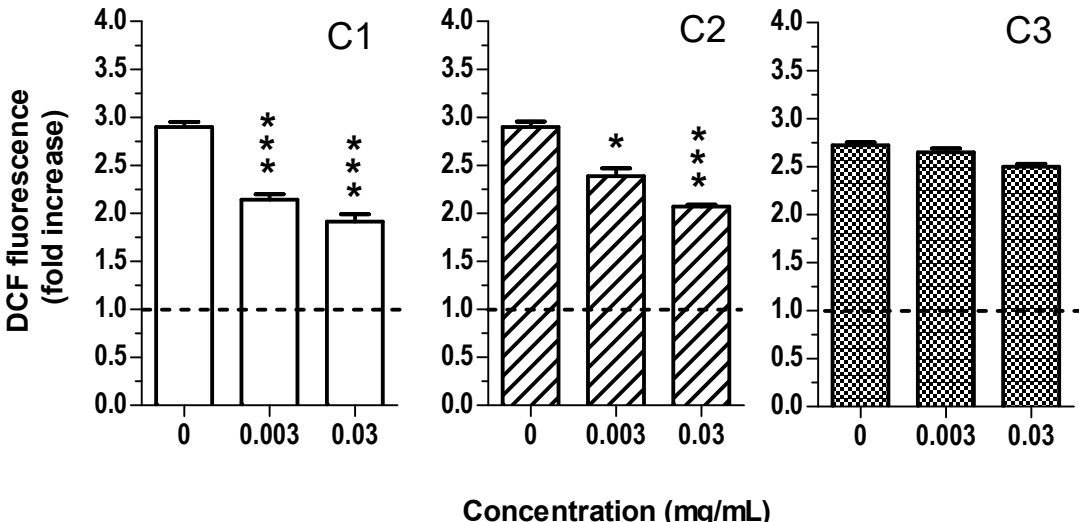

**Figure 5.** C1, C2 and C3 extracts reduce oxidative stress induced by UVA in HaCaT cells. Cells were treated with various concentrations of extract (0.003–0.03 mg/mL) for 1 h and then exposed to 5J/cm$^2$ of UVA. At the end of treatment, intracellular oxidative stress was evaluated using the probe DCFH$_2$-DA, as described in the method section. Data are expressed as fold increase of fluorescent DCF formation versus untreated cells and reported as mean $\pm$ SEM of four independent experiments (* $p < 0.05$ and *** $p < 0.001$ vs. cells treated with UVA; one-way ANOVA with Dunnett post hoc test).

Given the decreasing order of intrinsic antioxidant activity and total phenol content of C2 > C1 > C3 extracts, we suppose that the comparable activity of C1 and C2 in counteracting intracellular oxidative stress could be mainly ascribed to their similar flavonol levels. Probably, the higher anthocyanin levels found in C2 than C1 do not influence the intracellular antioxidant activity of these extracts. In this regard, several studies reported that anthocyanins have a low intracellular bioavailability compared to flavonols [42]. Moreover, our data are in agreement with the results of Menghini et al. [31] that demonstrated a significant antioxidant activity of water and ethanol extracts obtained by saffron waste products (anthers and tepals) in mouse myoblast C2C12 and human colon HCT116 cells. Moreover, regarding the molecular mechanism behind C1 and C2 ability to reduce oxidative stress, we hypothesize that it could be ascribed to a direct antioxidant mechanism. In fact, we evaluated the antioxidant activity after short treatment times (1 or 2 h), which are not compatible with the longer times required by an indirect antioxidant mechanism which involves the synthesis of new proteins. The ability of the extracts to directly counteract oxidant species in HaCaT cells was also supported by the data obtained with ABTS and DPPH assay.

By contrast to UVA, UVB radiation is less oxidant and more cytotoxic at keratinocyte level [43]. On these bases, we investigated the cytoprotective activity of the extracts against the cytotoxicity evoked by 50 mJ UVB (Figure 6). The treatment of HaCaT cells with C1 and C2 extracts for 1 h before the exposure with UVB significantly reduced the cytotoxicity confirming the photoprotective properties of these extracts.

Among the irritant compounds for the skin, the surfactants present in several cleansing formulations interact with the skin, inducing epidermal barrier injury, skin dryness, irritation, erythema, itching, and thickening [44]. In vitro studies, skin irritation induced by surfactants has been widely mimicked by exposing keratinocyte cells to sodium dodecyl sulfate (SDS) [44–46]. In order to evaluate the cytoprotective effects of the extracts against skin irritation, HaCaT cells were treated with the extracts (0.003 and 0.03 mg/mL) for 2 h, then treated with 140 µM SDS for 24 h (Figure 7). Interestingly, only C2 extract was able to significantly reduce cytotoxicity induced by SDS, suggesting that this cytoprotective effect cannot be explained in terms of different amount of flavonols and/or anthocyanins present in the studied extracts. The solvents, such as ethanol, water, and glycerine, used to

obtain C1, C2, and C3 extracts can extract different lipophilic or hydrophilic phytochemi-
cals. Among the hydrophilic phytochemicals found in C2 extract, the highest anthocyanin
levels could further contribute to cytoprotective effects of C2 extract at membrane level.
Anthocyanins are soluble in water and penetrate only the outer part of the external lipid
layer of the membrane, affecting its shape and lipid packing order as well as antioxidant
protection [29,47].

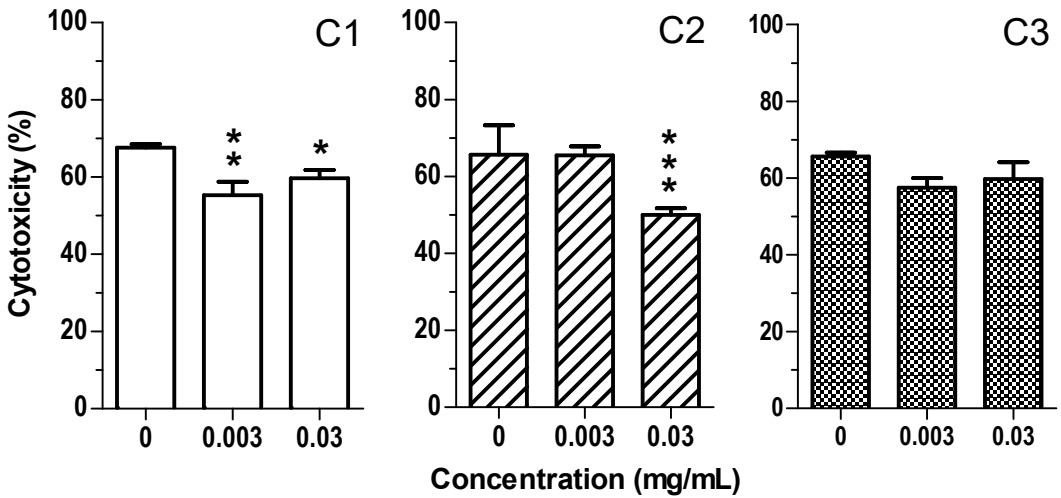

**Figure 6.** C1 and C2, but not C3, extracts reduce the cytotoxicity induced by UVB in HaCaT cells. Cells were treated
with various concentrations of extract (0.003–0.03 mg/mL) for 1 h and then exposed to 50 mJ/cm$^2$ of UVB. At the end of
treatment, cell viability was evaluated by MTT assay, as described in the method section. Data are expressed as percentage
of control cells and expressed as mean $\pm$ SEM of four independent experiments (* $p < 0.05$, ** $p < 0.01$ and *** $p < 0.001$ vs.
cells treated with UVA; one-way ANOVA with Dunnett post hoc test).

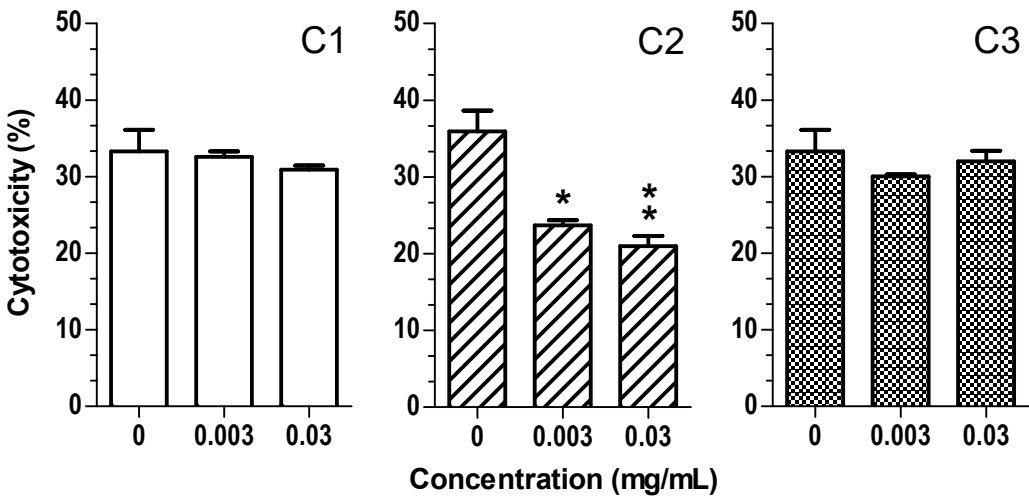

**Figure 7.** C2 extract reduces the cytotoxicity induced by SDS in HaCaT cells. Cells were treated with various concentrations
of extract (0.003–0.03 mg/mL) for 2 h and then treated with SDS (140 µM) for 24 h. At the end of treatment, cell viability
was evaluated by MTT assay, as described in the method section. Data are expressed as percentage of control cells and
expressed as mean $\pm$ SEM of four independent experiments (* $p < 0.05$, ** $p < 0.01$ vs. cells treated with UVA; one-way
ANOVA with Dunnett post hoc test).

A recent study investigated the effects of saffron floral biowaste in keratinocytes [48]
and demonstrated that a saffron flower acetone extract also promotes cell migration and
proliferation, suggesting pronounced wound healing properties. Taken together, these
results suggest a potential use of these biowaste materials as a source of cosmeceutical
compounds with multifunctional activity.

## 4. Conclusions

The main objective of this research project was to recycle *Crocus sativus* L. tepals, as spent material generated from the production of saffron spice, by applying several green extraction methods. The microwave-mediated green solvents extraction (MGSE) in water was selected as the best extraction method allowing to recover extracts with high content in flavonols and anthocyanins. The results obtained in product ion, used for the identification of flavonoids, highlighted the fundamental contribution of this work compared with the results already present in the literature. Here, we proposed a method in which the use of different collision energies has allowed us to detect in MS2 several numbers of fragment ion not found before. So, according to this approach, we successfully identified up to 27 flavonoids even with the HRMS analysis. Indeed, using UPLC-HRMS spectrometry, we tentatively identified for the first time two new molecular ions in *Crocus sativus* L. tepals. By correlating the fragments observed with a putative structure from database mass, we were able to identify two new molecular ions, the first one at $m/z$ 643.1507 that was assigned to the myricetin-di-glucoside, while the second one at $m/z$ 713.1930–711.1776 was correlated to primflasine.

The extracts obtained with water and ethanol showed the highest biological activity. In particular both of them counteracted oxidative stress in keratinocyte HaCaT cells, the water extract was more cytoprotective against UVB-induced damage and was also able to reduce the cytotoxicity induced by SDS. According to all the results, we can conclude that our extract should be a valid candidate for cosmetic formulations used to support anti-aging and/or anti-skin irritation.

**Supplementary Materials:** The following are available online at https://www.mdpi.com/article/10.3390/cosmetics8020051/s1, Table S1: The linearity, the sensitivity, accuracy, precision of the developed UPLC-DAD-ESI-MS method, Table S2: Identification of flavonols and anthocyanins in saffron extracts by UPLC-MS/MS analysis.

**Author Contributions:** Conceptualization, supervision, and draft writing, P.D.M. and A.T.; methodology, P.D.M., M.R.G., R.C., C.A., A.T. and G.L.; formal analyses and methodology, M.C., D.V.P., C.V., L.P. and M.R.G.; data validation and interpretation, M.R.G., M.C., A.T. and C.A.; review of the final draft and validation, P.D.M. and R.C.; funding acquisition, P.D.M. and A.T. All authors have read and agreed to the published version of the manuscript.

**Funding:** This work was partially supported by the European Commission of an H2020-MSCA-RISE-2016 award through the CHARMED project (grant number 734684) and an H2020-MSCA-RISE-2017 award through the CANCER project (grant number 777682).

**Acknowledgments:** Authors acknowledge Ilaria Scortichini for her support during the experimental work.

**Conflicts of Interest:** No potential conflict of interest was reported by any of the authors.

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
