# Peer review of "Development of New Extracts of Crocus sativus L. By-Product from Two Different Italian Regions as New Potential Active Ingredient in Cosmetic Formulations"

_cosmetics, doi:10.3390/cosmetics8020051_

Round 1

Reviewer 1 Report

All comments are presented in the attached file

Author Response

Authors acknowledge and appreciate the reviewer’s comments.

Reviewer 2 Report

this paper is OK,

it maybe interesting.

my basical qustion is what is "new".

I wonder what is "new", as cosmetics.

only new materials (componets) or new ingredients as they are, should not be enough,

it means "new effects", hopefully inovative effects for human, i think.

Author Response

(The authors gave the same response as above.)

Reviewer 3 Report

The presented manuscript is well constructed and provides a valuable insight into the possibilities of saffron flowers revalorization and their biological activity. The study is adequately designed, analysis carried out throughoutly and results well elaborated.

I only have few remarks regarding the manuscript itself.

In experimental section, authors should describe the extraction procedure more precisely as there is no dana given on the mass of flowers and volume of solvent used for the extraction. Furthermore, in the first part of the manuscript (abstract, introduction, methodology) authors mention saffron flowers without stigmas, while title indicates By-products as plural and in conclusion authors mention petals and sepals. This is a little confusing as it may indicate separate analysis of these segments, so I advise precise description of the material used (dried petals and sepals together or?).

Finally, I advise the English language revision by fluent speaker as some sentences have incorrect constructions and are difficult to understand.

Author Response

(The authors gave the same response as above.)

Round 2

Reviewer 1 Report

Thanks to the author for corrections and additional explanations. Please write "Crocus sativus" in italics. Your manuscript is very good. I recommend accept it for publication.